# A Surveillance Endoscopy Strategy Based on Local Recurrence Rates after Colorectal Endoscopic Submucosal Dissection

**DOI:** 10.3390/jcm10194591

**Published:** 2021-10-05

**Authors:** Jin Hwa Park, Ji Young Yoon, Sung Wook Hwang, Sang Hyoung Park, Dong-Hoon Yang, Byong Duk Ye, Seung-Jae Myung, Suk-Kyun Yang, Jeong-Sik Byeon

**Affiliations:** Asan Medical Center, Department of Gastroenterology, University of Ulsan College of Medicine, Olympic-ro 43-gil, Songpa-gu, Seoul 05505, Korea; pjh6718@hanmail.net (J.H.P.); fatei@hanmail.net (J.Y.Y.); snow903@gmail.com (S.W.H.); umdalpin@hanmail.net (S.H.P.); dhyang@amc.seoul.kr (D.-H.Y.); bdyemd@gmail.com (B.D.Y.); sjmyung@gmail.com (S.-J.M.); sky@amc.seoul.kr (S.-K.Y.)

**Keywords:** endoscopic submucosal dissection, recurrence, colorectal neoplasm, surveillance

## Abstract

Backgrounds: It is not clear when and how frequently surveillance endoscopy should be performed after colorectal endoscopic submucosal dissection (ESD). We aimed to suggest a surveillance endoscopy strategy by investigating the cumulative local recurrence rates and identifying risk factors for local recurrence after colorectal ESD. Methods: We reviewed the medical records of 770 patients who underwent colorectal ESD for 778 lesions at our institution from 2005 to 2016. We investigated the cumulative local recurrence rates and risk factors for local recurrence. Results: Local recurrence developed in 12 (1.5%) of 778 lesions during the follow-up period of 37.4 ± 31.7 months. The one-, three-, and five-year cumulative local recurrence rates were 0.4%, 1.7%, and 2.2%, respectively. The risk factors for local recurrence were piecemeal resection (odds ratio (OR) 3.948, 95% confidence interval (CI) 1.164–13.385; *p* = 0.028) and histological incomplete resection (OR 8.713, 95% CI 2.588–29.334; *p* < 0.001). Local recurrence tended to develop frequently after ESD of early cancers. Conclusions: Short-term surveillance endoscopy should be recommended after piecemeal ESD, histological incomplete resection, and ESD of early colorectal cancers. Surveillance endoscopy with longer intervals can be suggested after en bloc ESD with the histological complete resection of benign colorectal tumors.

## 1. Introduction

Colorectal cancer is one of the most common cancers worldwide and the third cause of cancer-related deaths [1]. As most colorectal cancers develop through the adenoma-carcinoma sequence [2], screening for colorectal adenoma and/or early colorectal cancer and their endoscopic resection can prevent the progression to advanced cancer and avoid colorectal cancer-related mortality [3,4]. 

Endoscopic submucosal dissection (ESD) is one of the several endoscopic resection techniques for colorectal neoplasms. Despite a relatively high rate of adverse events including perforation, ESD is considered a good treatment option particularly in large laterally spreading tumors (LSTs) and early colorectal cancers, including mucosal and superficial submucosal cancer, because of its high en bloc resection rate, even in large lesions [5,6,7,8].

After endoscopic resection of colorectal neoplasms, surveillance endoscopy should be performed regularly to manage local recurrence and metachronous neoplasms. Several international guidelines recommend strategic endoscopy surveillance intervals according to the risk of local recurrence and metachronous neoplasms [9,10]. However, these intervals are based on previous studies that investigated mainly conventional endoscopic resection techniques, such as snare polypectomy, endoscopic mucosal resection, and endoscopic piecemeal mucosal resection. 

Although there are several studies evaluating the frequency of and risk factors for local recurrence after colorectal ESD [7,11,12,13,14], only a few studies have investigated yearly cumulative recurrence rates. Due to a lack of robust data on cumulative recurrence, the current guidelines do not specifically suggest appropriate surveillance endoscopy intervals after colorectal ESD. Thus, post-ESD surveillance endoscopy intervals are generally recommended at the discretion of individual endoscopists. The absence of a standardized surveillance endoscopy strategy may hinder the effective monitoring of local recurrence and worsen the cost-effective approach for colorectal ESD. Therefore, it is of paramount importance to establish a standardized endoscopy surveillance strategy based on the systematic analysis of robust colorectal ESD data.

The purpose of this study was to investigate the frequency of local recurrence and the cumulative local recurrence rate after colorectal ESD. We also aimed to investigate the risk factors for local recurrence after colorectal ESD and stratify the risk of cumulative recurrence according to the risk factors. Finally, we aimed to suggest an appropriate surveillance endoscopy strategy after colorectal ESD.

## 2. Materials and Methods

### 2.1. Study Design

This study was a retrospective review of the medical records of 1281 patients who underwent colorectal ESD for 1293 lesions at Asan Medical Center, Seoul, from January 2005 to December 2016. Of 1293 colorectal ESD cases, 515 were excluded because of loss to follow-up, post-ESD subsequent colorectal surgery, and ESD for subepithelial lesions (Figure 1). A total of 770 patients with 778 colorectal lesions removed by ESD were included in the final analysis. 

We investigated the baseline characteristics of the enrolled patients, short-term outcomes of colorectal ESD, and long-term courses, including local recurrence, by reviewing medical records and endoscopy images. The first surveillance endoscopy after colorectal ESD was performed usually 1 year after en bloc resection and 6 months after piecemeal resection. Subsequent surveillance endoscopy intervals were decided according to the findings of the first surveillance endoscopy. Surveillance endoscopy intervals could be modified if clinically indicated. The protocol of this study was approved by the institutional review board of our institution (2017-0756).

### 2.2. Colorectal ESD Procedures and Histopathological Examination

Colorectal ESD in this study included two types of ESD procedures: ESD throughout and hybrid ESD [15,16]. ESD throughout was defined as conventional ESD in which submucosal dissection was performed until the complete resection of a tumor. Hybrid ESD was defined as partial ESD with final snaring. Indications of hybrid ESD were (1) the rescue procedure in the case of difficulty of ESD throughout and (2) a method for rapid completion of the resection procedure if effective snaring for complete resection looked easy and secure after sufficient submucosal dissection. The procedural details of colorectal ESD were described in our previous report [15]. 

The underwater technique was not used in this study. All ESD procedures were performed by board-certified gastrointestinal endoscopists with previous experience of therapeutic colonoscopy procedures, including endoscopic mucosal resection of large polyps for >5 years. En bloc resection was defined as the excision of a tumor in one piece. The resected specimen was stretched, pinned, fixed in formalin, and evaluated under a microscope. The World Health Organization classification for tumors of the digestive system was used for histopathological evaluation [17]. 

Histological complete resection was defined as en bloc resection with negative lateral and deep resection margins [18]. Curative resection in an early colorectal cancer was defined as histological complete resection with no risk of lymph node metastasis according to the histological criteria given by the Japanese Society for Cancer of the Colon and Rectum guidelines [19]. Although post-ESD subsequent surgery was performed in most patients with early colorectal cancer in whom curative resection was not achieved, some patients who refused surgery were followed up without surgery.

### 2.3. Definition of Terms

The morphology of colorectal neoplasms was classified into polypoid and non-polypoid tumors according to the Paris classification. As no pedunculated tumors were resected by ESD, all polypoid lesions were classified as type I. All non-polypoid tumors could be regarded as LSTs, which were classified into granular LSTs (homogenous (LST G-H) and nodular mixed (LST G-NM)) and non-granular LSTs (flat elevated (LST NG-FE) and pseudodepressed (LST NG-PD)) [20,21]. The location of a colorectal lesion was classified as either colonic or rectal. 

Colonic location was classified as either the right colon (cecum, ascending colon, and proximal two-thirds of the transverse colon) or the left colon (distal one-third of the transverse colon, descending colon, and sigmoid colon). ESD time was defined as the time from submucosal injection to the end of resection. We used indigo carmine-mixed solution as a submucosal injection fluid, which could stain the loose submucosal layer blue but not the fibrotic scar tissue. The degree of submucosal fibrosis was classified as either absence or presence. 

The absence of fibrosis was defined as no fibrosis manifesting as a blue transparent layer after submucosal injection due to the absence of unstained white scar fibers. The presence of fibrosis was defined as when the submucosa appeared as a white web-like structure in the blue submucosal layer or as a white muscular structure without a blue transparent layer in the submucosal layer because of the presence of unstained white scar fibers [22]. 

Delayed bleeding was defined as hematochezia or melena requiring an endoscopic hemostatic procedure after ESD completion. Perforation was diagnosed endoscopically and/or radiologically. Local recurrence was histologically defined when a biopsy of an apparent or suspected lesion at the ESD site during a surveillance endoscopy showed an adenoma or adenocarcinoma.

### 2.4. Statistical Analysis

Categorical and nominal variables are expressed as numbers with percentages, and continuous variables are presented as means (standard deviations). Analysis of variance, Student’s t-test, and the chi-squared test were used to examine differences among the groups. The cumulative recurrence rates and risk factors for local recurrence were investigated by using Kaplan–Meier curves and logistic regression analysis. Odds ratios (ORs) and 95% confidence intervals (CIs) were calculated for each variable. All reported *p*-values were two-tailed, and the significance level was set at 0.05. Statistical analyses were performed using Microsoft Office Excel 2010 (Microsoft, Redmond, WA, USA) and SPSS Statistics for Windows, version 24.0 (IBM, Armonk, NY, USA).

## 3. Results

### 3.1. Baseline Characteristics of Patients and Colorectal Lesions

The mean age of the patients was 61.4 years, and 452 were male (58.7%). The rectum was the most common lesion site (352/778, 45.2%). LST G-NM was the most common morphological type (38.0%) followed by Is, LST NG-FE, LST NG-PD, and LST G-H. The average size of lesions was 27.7 mm. A tubular or tubulovillous adenoma was the most common histology (56.1%), followed by mucosal cancer (30.7%) and submucosal cancer (10.9%). Detailed baseline characteristics are presented in Table 1.

ESD throughout and hybrid ESD were performed in 632 (81.2%) and 146 colorectal lesions (18.8%), respectively. The average duration of ESD procedures was 54.2 (43.6) min. En bloc resection was achieved in 688 lesions (88.4%). Histological complete resection was confirmed in 627 lesions (80.6%). Perforation occurred in 37 lesions (4.8%). All perforation cases were successfully managed with endoscopic clipping and antibiotics without surgery. Delayed bleeding occurred in 18 patients (2.3%), all of whom were managed with endoscopic treatment, such as clipping and hemostatic forceps (Table 2).

### 3.2. Local Recurrence after Colorectal ESD

The mean number of surveillance endoscopies was 2.3 (1.5) during the mean follow-up period of 37.4 (31.7) months. Local recurrence developed in 12 of 778 lesions (1.5%). The 1-, 3-, and 5-year local recurrence rates were 0.4%, 1.7%, and 2.2%, respectively (Figure 2). The 1-, 3-, and 5-year local recurrence rates were higher after piecemeal ESD than after en bloc ESD (2.5%, 6.3%, and 6.3% vs. 0.2%, 1.2%, and 1.7%, respectively; *p* = 0.048), and higher in cases of histological incomplete resection than in cases of histological complete resection (2.2%, 7.3%, and 7.3% vs. 0%, 0.5%, and 1.1%, respectively; *p* < 0.001) (Figure 3).

### 3.3. Risk Factors for Local Recurrence

Local recurrence was significantly more common after piecemeal ESD than after en bloc ESD (OR 3.948, 95% CI 1.164–13.385; *p* = 0.028). Local recurrence was more frequent in cases of histological incomplete resection than in those of histological complete resection (OR 8.713, 95% CI 2.588–29.334; *p* < 0.001). Other factors, such as age, sex, tumor size, tumor morphology, tumor location, fibrosis, procedure time, histology, and differentiation of adenocarcinoma, did not show any significant correlation with local recurrence (Table 3).

The clinical characteristics of 12 patients who showed local recurrence are summarized in Table 4. 

Seven of these patients showed histological incomplete resection with positive resection margins, three had piecemeal resection, and five showed mucosal or submucosal cancer. Only 2 of 12 patients with local recurrence did not show any of these findings; that is, their ESD results were en bloc ESD with histological complete resection of benign colorectal tumors. Local recurrences in these two patients were detected at 48 and 71 months after colorectal ESD, whereas local recurrences in other patients with risk factors for recurrence and/or early cancer were detected within 18 months after ESD.

## 4. Discussion

In this surveillance endoscopy study, we found that the local recurrence rate after colorectal ESD was 1.5%. Piecemeal resection and histological incomplete resection increased the risk of local recurrence. Some patients with local recurrence showed mucosal or submucosal cancer in their ESD specimens. On the basis of these findings, we suggest more frequent surveillance endoscopies after colorectal ESD in cases of piecemeal resection, histological incomplete resection, and early cancer, whereas a surveillance endoscopy strategy similar to that after conventional colonoscopic polypectomy could be recommended after colorectal ESD without these risk factors.

Several previous studies investigated the incidence of and risk factors for local recurrence after colorectal ESD. In a meta-analysis including 101 colorectal ESD studies, the 1-year cumulative recurrence rate was 2.0% [23]. Although this meta-analysis showed the 1-year local recurrence rate, it did not show detailed risk factors for local recurrence and could not suggest an appropriate surveillance endoscopy strategy based on cumulative local recurrence rates. A Japanese multi-center study analyzed the local recurrence after endoscopic resection of 1845 large colorectal lesions in 1524 patients. The overall local recurrence rate was 4.3%. 

Piecemeal resection was a risk factor for local recurrence [14]. Although this study enrolled a large number of patients, both ESD and conventional endoscopic resection methods, such as endoscopic mucosal resection and snare polypectomy were performed. In addition, neither the follow-up period nor yearly cumulative recurrence rates were reported. Thus, the cumulative local recurrence after colorectal ESD and its risk factors could not be clearly determined. Another Japanese study observed 423 patients after colorectal ESD for 4.9 years [24]. The 3- and 5-year cumulative local recurrence rates were 2.9% and 3.8%, respectively. 

Piecemeal resection and submucosal deep tumor invasion were associated with the local recurrence. A Chinese study that observed 514 patients after ESD of 520 colorectal lesions for 58 months showed local recurrence in four (0.8%) patients. Piecemeal resection was a risk factor for local recurrence [25]. Although these two studies showed local recurrences and risk factors similar to our study, the number of patients analyzed was smaller when compared with our study. 

Thus, taking the findings of previous studies together with the findings of our present study, which performed more systematic analyses on cumulative local recurrence rates and risk factors in a larger number of patients, we suggest that the 1-, 3-, and 5-year cumulative local recurrence rates may be 0.4–2.0%, 1.7–2.9%, and 2.2–3.8%, respectively. We also suggest that piecemeal resection, histological incomplete resection, and early cancer may be risk factors for local recurrence after colorectal ESD.

In our study, piecemeal resection and histological incomplete resection were significant risk factors for local recurrence. Previous studies also suggested that piecemeal resection [14,23,24,25] and histological incomplete resection [24] might increase the risk of local recurrence after colorectal ESD. We believe that piecemeal resection and histological incomplete resection are two important risk factors for local recurrence after colorectal ESD with high confidence on the basis of consistent findings between our current study and previous studies.

Although not statistically significant in our study, local recurrence tended to increase after ESD of early colorectal cancer, especially submucosal cancer (OR 2.648, 95% CI 0.649–10.804). In addition, 5 of 12 patients with local recurrence showed early colorectal cancer in their ESD specimens. A previous study showed an increase in local recurrence in the presence of submucosal tumor invasion, particularly with deep submucosal invasion greater than SM2 [24]. 

Although the reason for the high risk of local recurrence after ESD of early colorectal cancer is not yet clear, a previous study suggested that invasive procedures, such as a biopsy and polypectomy, might lead to viable cancer-cell seeding through endoscopic accessory devices and the endoscopic working channel [26]. Colorectal ESD requires a longer procedure time and many more repeated ESD knife passages through the working channel than a biopsy or conventional polypectomy. 

This more complicated nature of colorectal ESD may be accompanied by a higher risk of implanting cancer cells in the ESD ulcer bed, which can partly explain the high risk of local recurrence after the colorectal ESD of early cancer. Another issue regarding early cancer and non-curative resection in our study is the completeness of histological reports. We did not perform histological re-examination of the resected specimens. As consistent re-review of the pathology slides may improve the quality of histological examination, future studies should perform histological re-examination to clearly analyze the risk of early cancer and non-curative resection.

Interestingly, local recurrence after colorectal ESD was found within 18 months in patients with risk factors, such as piecemeal resection, histological incomplete resection, and/or early cancer. In comparison, local recurrence was detected at 48 and 71 months in two patients without these risk factors. This trend was similar to a previous study that reported local recurrence at 4–18 months after piecemeal ESD and/or colorectal ESD with histological incomplete resection and local recurrence at 59 and 74 months after colorectal ESD without these risk factors [24]. Considering these findings, initial surveillance endoscopies at short intervals should be recommended after colorectal ESD in cases of piecemeal resection, histological incomplete resection, and/or early cancer.

On the basis of our findings and the current international guidelines for surveillance colonoscopy after a polypectomy [9,10,27], we proposed specific recommendations for surveillance endoscopy after colorectal ESD in Table 5. We suggest the first surveillance endoscopy at 3 years after en bloc ESD with histological complete resection of a benign tumor based on the very low risk of local recurrence and relatively long latency period before the detection of benign recurrence in these cases. 

The timing of second surveillance endoscopy may be determined according to the findings of the first surveillance endoscopy. We suggest the first surveillance endoscopy at 6 months after piecemeal ESD and histological incomplete resection and the second endoscopy at 1 year after the first surveillance based on the high risk of local recurrence and tendency of early recurrence in these circumstances. 

We suggest the first surveillance endoscopy at 3–6 months after ESD of early colorectal cancer, and surveillance should be repeated every 3–6 months for 2–3 years based on a relatively high risk of local recurrence in early colorectal cancer in our study and previous studies. Another reason for the recommendation of short-interval surveillance endoscopy after colorectal ESD of early cancer is the difficulty in endoscopic re-treatment in cases of late detection of cancer recurrence with a resultant poor prognosis, unlike the recurrence of benign adenomas that can be mostly treated by repeat endoscopic resection [28].

Our current study had several limitations. First, this was a retrospective study using medical records. Thus, the surveillance endoscopy intervals were not consistent in all patients and were subjectively determined by the physician in charge, which may have biased the outcome of the analysis. Second, the average follow-up period was not long enough for more confirmative analyses. Third, although the number of enrolled patients was relatively large, there were only 12 patients with local recurrence, making it difficult to analyze the risk factors in more detail. 

In addition, we could not perform statistically meaningful analyses in adenoma, mucosal cancer, and submucosal cancer groups separately because of the small number of cancers, especially, submucosal cancers. Future studies should investigate the risk factors for recurrence in cancer and non-cancer separately because they are completely different in nature and have different prognoses. 

Finally, this study was a single-center analysis of a limited number of patients, which compromises the generalizability of our findings. Further large scale, multi-center studies should be performed to achieve confirmative conclusion by statistically powerful analyses. Despite these limitations, we believe that our study is meaningful as, to the best of our knowledge, we proposed the first surveillance endoscopy recommendation based on cumulative local recurrence rates and risk factors for local recurrence after colorectal ESD.

## 5. Conclusions

In conclusion, our study determined the 1-, 3-, and 5-year cumulative local recurrence rates after colorectal ESD to be 0.4%, 1.7%, and 2.2%, respectively. We suggest that surveillance endoscopy may be performed within 6 months after colorectal ESD in the presence of risk factors, such as piecemeal resection, histological incomplete resection, and early cancer. Surveillance endoscopy may be performed at longer intervals after colorectal ESD of benign lesions without risk factors.

## Figures and Tables

**Figure 1 jcm-10-04591-f001:**
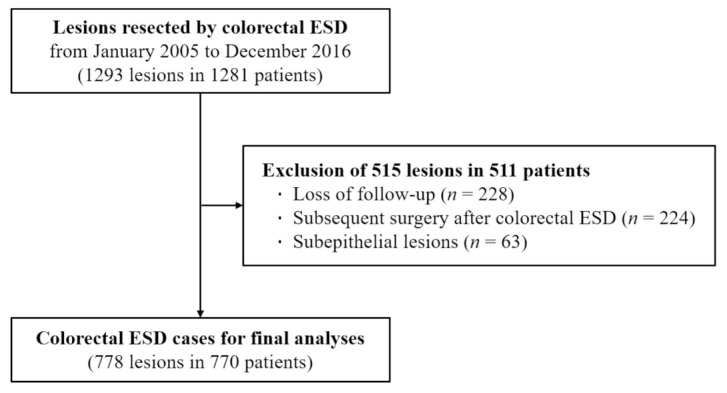
Flow chart for inclusion of patients.

**Figure 2 jcm-10-04591-f002:**
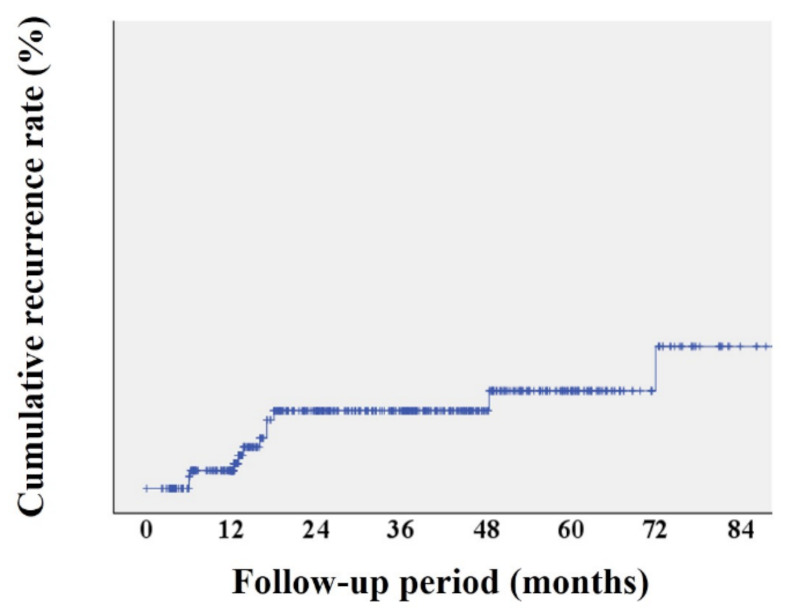
Overall cumulative recurrence after colorectal endoscopic submucosal dissection.

**Figure 3 jcm-10-04591-f003:**
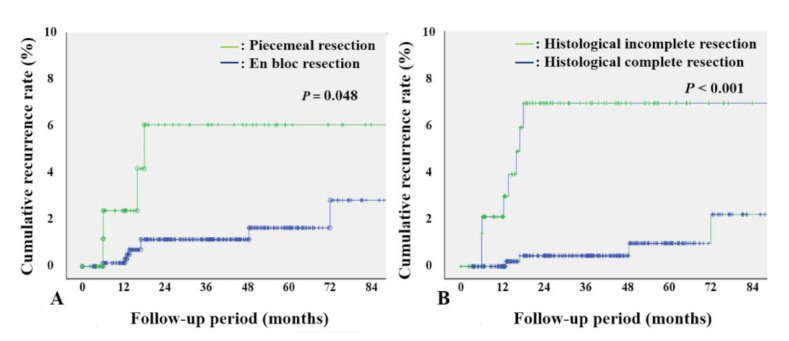
Cumulative recurrence rates according to risk factors. (**A**) Cumulative recurrence rates after piecemeal and en bloc endoscopic submucosal dissection. (**B**) Cumulative recurrence rates after histological complete and incomplete resection.

**Table 1 jcm-10-04591-t001:** Baseline characteristics of the patients and colorectal lesions.

Characteristics	
Age (year)	61.4 ± 9.9
Sex (%)	
Male	452 (58.7%)
Female	318 (41.3%)
Tumor location	
Right colon	276 (35.5%)
Left colon	150 (19.3%)
Rectum	352 (45.2%)
Tumor morphology (%)	
LST-G-H	77 (10.0%)
LST-G-NM	296 (38.0%)
LST-NG-FE	144 (18.5%)
LST-NG-PD	109 (14.0%)
Is	152 (19.5%)
Tumor size (mm)	27.7 ± 14.1
Histology (%)	
Adenoma (tubular or tubulovillous)	436 (56.1%)
Sessile serrated lesion	18 (2.3%)
Adenocarcinoma (mucosal cancer)	239 (30.7%)
Adenocarcinoma (submucosal cancer)	85 (10.9%)
Differentiation of adenocarcinoma (%)	
Well-differentiated	265 (81.8%)
Moderately differentiated	58 (17.9%)
Poorly differentiated	1 (0.3%)

LST-G-H, granular homogenous laterally spreading tumor; LST-G-NM, granular nodular mixed laterally spreading tumor; LST-NG-FE, non-granular flat elevated laterally spreading tumor; LST-NG-PD, non-granular pseudodepressed laterally spreading tumor.

**Table 2 jcm-10-04591-t002:** Colorectal ESD procedure-related outcomes.

Characteristics	
Type of ESD (%)	
ESD throughout	632 (81.2%)
Hybrid ESD	146 (18.8%)
Submucosal fibrosis (%)	161 (20.7%)
En bloc resection	
Yes	688 (88.4%)
No (piecemeal resection)	90 (11.6%)
ESD time (min)	54.2 ± 43.6
Adverse events (%)	
Perforation	37 (4.8%)
Delayed bleeding	18 (2.3%)
Deep margin involvement (%)	8 (1.0%)
Lateral margin involvement (%)	89 (11.4%)
Histological complete resection (%)	627 (80.6%)
Follow-up duration (months)	37.4 ± 31.7
Frequency of surveillance endoscopy	2.3 ± 1.5
Local recurrence (%)	12 (1.5%)

ESD, endoscopic submucosal dissection.

**Table 3 jcm-10-04591-t003:** Risk factors of local recurrence after colorectal ESD.

	OR (95% CI)	*p*
Age	1.005 (0.948–1.065)	0.861
Sex		
Male	1.000	
Female	0.702 (0.224–2.197)	0.544
Tumor size	1.002 (0.963–1.042)	0.937
Tumor location		
Rectum	1.000	
Right colon	1.846 (0.473–7.207)	0.377
Left colon	1.230 (0.203–7.442)	0.822
Tumor morphology		
Is	1.000	
LST G-H	0.387 (0.044–3.371)	0.390
LST G-NM	0.301 (0.071–1.277)	0.103
LST NG-FE	0.414 (0.079–2.169)	0.297
LST NG-PD	0.272 (0.031–2.364)	0.238
Submucosal fibrosis		
Absence	1.000	
Presence	2.293 (0.663–7.934)	0.190
Procedure time	1.004 (0.996–1.013)	0.340
*En bloc* resection		
Yes	1.000	
No (piecemeal resection)	3.948 (1.164–13.385)	0.028
Histology		
Adenoma	1.000	
Sessile serrated lesion	1.013 (0.995–1.073)	0.999
Mucosal cancer	0.909 (0.225–3.667)	0.893
Submucosal cancer	2.648 (0.649–10.804)	0.175
Differentiation of adenocarcinoma		
Well-differentiated	1.000	
Moderately differentiated	2.260 (0.403–12.666)	0.354
Poorly differentiated		1.000
Histological complete resection		
Yes	1.000	
No	8.713 (2.588–29.334)	<0.001

ESD, endoscopic submucosal dissection; LST-G-H, granular homogenous laterally spreading tumor; LST-G-NM, granular nodular mixed laterally spreading tumor; LST-NG-FE, non-granular flat elevated laterally spreading tumor; LST-NG-PD, non-granular pseudodepressed laterally spreading tumor.

**Table 4 jcm-10-04591-t004:** Clinical characteristics of 12 patients with local recurrence.

Patient Number	Sex/Age	Tumor Size (mm)	Tumor Location	Tumor Morphology	Risk Factors for Local Recurrence	Histology	Time of Recurrence after Colorectal ESD (Months)
Patient 1	M/76	25	Rectum	LST NG-PD	Histological incomplete resection	Submucosal cancer	6
Patient 2	M/54	20	Rectum	Is	None	Benign	48
Patient 3	F/64	33	Right colon	LST NG-FE	None	Benign	71
Patient 4	M/55	30	Left colon	Is	Histological incomplete resection, piecemeal resection	Benign	6
Patient 5	M/50	15	Rectum	LST NG-FE	None	Submucosal cancer	17
Patient 6	F/67	56	Rectum	LST G-NM	None	Mucosal cancer	17
Patient 7	F/53	35	Left colon	Is	Histological incomplete resection	Benign	12
Patient 8	F/78	33	Left colon	LST G-NM	Histological incomplete resection	Benign	16
Patient 9	M/66	32	Right colon	LST G-NM	Histological incomplete resection, piecemeal resection	Benign	6
Patient 10	M/58	30	Right colon	LST NG-FE	Histological incomplete resection, piecemeal resection	Benign	18
Patient 11	F/49	25	Rectum	LST G-H	None	Mucosal cancer	13
Patient 12	M/73	20	Rectum	Is	Histological incomplete resection	Submucosal cancer	13

**Table 5 jcm-10-04591-t005:** Proposal of surveillance endoscopy strategies after colorectal ESD.

	First Surveillance	Second Surveillance
En bloc ESD with histological complete resection of benign tumors ^1^	3 years after ESD	Interval adjustment based on the findings of the first surveillance endoscopy
Piecemeal ESD or histological incomplete resection of benign tumors ^2^	6 months after ESD	One year after the first surveillance endoscopy
ESD of early colorectal cancer ^3^	3–6 months after ESD	Repeat surveillance at 3–6-month intervals for 2–3 years

^1^ Based on the findings of our study and the current USMSTF and ESGE guidelines [9,10]. ^2^ Based on the findings of our study ^3^ Based on the findings of our study and the current USMSTF guideline [27]. ESD, endoscopic submucosal dissection; USMSTF, US Multisociety Task Force on Colorectal Cancer; ESGE, European Society of Gastrointestinal Endoscopy.

## Data Availability

The data presented in this study are available on request from the corresponding author. The data are not publicly available due to contain patient personal information.

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
