# Peer review of "A Surveillance Endoscopy Strategy Based on Local Recurrence Rates after Colorectal Endoscopic Submucosal Dissection"

_jcm, 2021, doi:10.3390/jcm10194591_

Round 1
Reviewer 1 Report
The authors discuss how frequently surveillance endoscopy should be performed after colorectal endoscopic submucosal dissection (ESD). They investigate cumulative local recurrence rates and identifying risk factors for local recurrence after colorectal ESD in Korean Registry. They concluded that short-term surveillance endoscopy should be recommended after piecemeal ESD, incomplete histological resection, and ESD of early colorectal cancers. It is a well-written article and is an essential aspect of this field of study. However, there are some concerns of this article.
Minor:
- Line 120-123 Absence of fibrosis was defined ~ Please explain the word of “ a blue transparent layer.”
- Would you please explain the detail of performers of ESD, such as skills?
- In hybrid ESD, Did you planned before ESD? Or, only in case of difficulty of ESD throughout?
- In the resection of LST, Did you consider the underwater technique?
Author Response
- Line 120-123 Absence of fibrosis was defined ~ Please explain the word of “a blue transparent layer.”
☞ Thank you for your good comment.
The solution used for submucosal injection contains indigo carmine, which makes the injection solution blue. Once the blue injection solution is injected to the normal submucosal layer without fibrosis, it is changed to a blue transparent layer because of the absence of fibrotic tissue composed of scar fibers. However, if there is fibrosis in the submucosal layer, it looks like a white web-like structure rather than a blue transparent layer after injection of blue indigo carmine solution because the fibrotic tissue is composed of thick white scar fibers which is not completely stained by the blue indigo carmine solution.
Thus, we can evaluate the presence or absence of fibrosis according to the appearance of the submucosal layer after submucosal injection of blue indigo carmine solution.
We clearly revised related parts in the method section of our revised manuscript as follows.
“We used indigo carmine-mixed solution as a submucosal injection fluid, which could stain the loose submucosal layer blue but not the fibrotic scar tissue. The degree of submucosal fibrosis was classified as either absence or presence. Absence of fibrosis was defined as no fibrosis manifesting as a blue transparent layer after submucosal injection because of absence of unstained white scar fibers. Presence of fibrosis was defined as when the submucosa appeared as a white web-like structure in the blue submucosal layer or as a white muscular structure without a blue transparent layer in the submucosal layer because of presence of unstained white scar fibers.”
- Would you please explain the detail of performers of ESD, such as skills?
☞ All ESD procedures were performed by staff endoscopists at our institution who were board-certified gastrointestinal endoscopists with previous experience of therapeutic colonoscopy procedures including endoscopic mucosal resection of large polyps for > 5 years. We added this description in the method section of the revised manuscript as follows.
“All ESD procedures were performed by board-certified gastrointestinal endoscopists with previous experience of therapeutic colonoscopy procedures including endoscopic mucosal resection of large polyps for > 5 years.”
- In hybrid ESD, Did you planned before ESD? Or, only in case of difficulty of ESD throughout?
☞ Thank you for this comment.
We performed hybrid ESD in case of difficulty of ESD throughout. Another indication of hybrid ESD was for rapid completion of the resection procedure if effective snaring for complete resection looked easy and secure after sufficient submucosal dissection. We added the following description in the method section of the revised manuscript.
“Indications of hybrid ESD were 1) the rescue procedure in case of difficulty of ESD throughout and 2) a method for rapid completion of the resection procedure if effective snaring for complete resection looked easy and secure after sufficient submucosal dissection.”
- In the resection of LST, Did you consider the underwater technique?
☞ No, we did not consider the underwater technique. The underwater technique was proposed in 2012 – 2016. In comparison, the subjects of this study were patients who underwent ESD between 2005 and 2016. Thus, most patients underwent the resection procedure before introduction of underwater technique. We added the following brief comment in the method section of our revised manuscript.
“Underwater technique was not used in this study.”
Reviewer 2 Report
This paper is a retrospective study of the recurrence rate and risk factors for local recurrence after colorectal ESD, based on data from a single institution. The following is a list of improvements to this paper.
1. Originally, cancer and adenoma are completely different in nature, and cancer and non-cancer should be discussed separately. Furthermore, mucosal and submucosal cancers should also be discussed separately. The fundamental problem with this article is that it analyzes adenoma, mucosal cancer, and submucosal cancer all together. Therefore, the conclusion that non-curative resection, segmental resection, and cancer cases are risk factors for recurrence is quite obvious.
2. This is a retrospective, single-center study with too few cases. How do you calculate the power with regard to case accumulation?
3. The results show that in cases of colorectal ESD for cancer (adenocarcinoma: mucosal cancer 239, submucosa cancer 85, total 324 cases), curative resection was performed. Three of the cases that were judged to be curative resections were recurrences, especially two cases of mucosal cancer that had no risk factors of recurrence. If non-curative resections are excluded, the recurrence rate would be even higher. For example,the pathological diagnosis of the resected specimens need to be re-examined.
4. Because this was a retrospective study, the surveillance intervals were subjectively determined by the physician in charge, which may have biased the outcome of the analysis.
Author Response
- Originally, cancer and adenoma are completely different in nature, and cancer and non-cancer should be discussed separately. Furthermore, mucosal and submucosal cancers should also be discussed separately. The fundamental problem with this article is that it analyzes adenoma, mucosal cancer, and submucosal cancer all together. Therefore, the conclusion that non-curative resection, segmental resection, and cancer cases are risk factors for recurrence is quite obvious.
☞ Thank you for your good comment.
We completely agree to your opinion that the analyses should have been made separately according to the histologic results of resected specimens. Nonetheless, we could not perform statistically meaningful analyses separately in each histology group because of the small number of cancer cases, especially, submucosal cancers. Thus, we added this point as a limitation of our study in the revised discussion section as follows.
“In addition, we could not perform statistically meaningful analyses in adenoma, mucosal cancer, and submucosal cancer groups, separately, because of the small number of cancers, especially, submucosal cancers. Future studies should investigate risk factors for recurrence in cancer and non-cancer, separately, because they are completely different in nature and have different prognosis.”
As you pointed out, the risk of recurrence is different between cancer and adenoma because of their different nature. Thus, the surveillance interval should also be different according to the histology. Despite this obviousness, to date, few studies suggested an appropriate surveillance interval according to the presence or absence of risk factors including histologic findings. We suggest our study was meaningful in terms that we not only identified obvious risk factors for recurrence, but also presented an appropriate surveillance interval after colorectal ESD of all histological lesions.
- This is a retrospective, single-center study with too few cases. How do you calculate the power with regard to case accumulation?
☞ Thank you for your important comment.
Although the number of cases (n=778) in our study was not small compared to those of previous studies, confirmative conclusion could not be made because of small number of local recurrence (n=12) and limitation of generalizability because of a single-center analysis of our study. Thus, we totally agree to your opinion and added the following description as a limitation in the discussion section of the revised manuscript.
“Finally, this study was a single-center analysis of limited number of patients, which compromises the generalizability of our findings. Further large scale, multi-center studies should be performed to achieve confirmative conclusion by statistically powerful analyses.”
- The results show that in cases of colorectal ESD for cancer (adenocarcinoma: mucosal cancer 239, submucosa cancer 85, total 324 cases), curative resection was performed. Three of the cases that were judged to be curative resections were recurrences, especially two cases of mucosal cancer that had no risk factors of recurrence. If non-curative resections are excluded, the recurrence rate would be even higher. For example, the pathological diagnosis of the resected specimens need to be re-examined.
☞ Thank you for your comment.
Because post-ESD subsequent surgery was performed in most patients with non-curative resection, only a few patients who refused surgery despite non-curative resection were followed up without surgery and included in our study. For example, one patient with poorly differentiated adenocarcinoma (Table 1) and 8 patients with deep margin involvement (Table 2) were included. However, these numbers could be changed if re-examination of histopathology of resected specimens are performed as you pointed out. Although we agree to your opinion that re-examination of histological findings may improve the quality of the study thereby achieving more confirmative conclusion, it was impossible because pathological slides of some patients who underwent ESD more than 10 years ago like 2005 – 2010 were not stored in our institution. Therefore, we added the following description in our revised discussion to address the reviewer’s comment.
“Another issue about early cancer and non-curative resection in our study is the completeness of histological reports. We did not perform histological re-examination of resected specimens. Because consistent re-review of the pathology slides may improve the quality of histological examination, future studies should perform histological re-examination to clearly analyze the risk of early cancer and non-curative resection.”
- Because this was a retrospective study, the surveillance intervals were subjectively determined by the physician in charge, which may have biased the outcome of the analysis.
☞ Thank you for your valuable comment.
We agree with you. Thus, we already described this point as a limitation in our original manuscript as follows.
“First, this was a retrospective study using medical records. Thus, the surveillance endoscopy intervals were not consistent in all patients.”
Nonetheless, we suggest the degree of difference in the surveillance interval between endoscopists was not large because the first surveillance endoscopy was performed usually one year after en bloc resection and 6 months after piecemeal resection in most cases as described in the method section. Anyhow, because we basically agree to the reviewer’s opinion on the risk of bias, we changed our original description above to the following sentence in the discussion section of the revised manuscript as follows.
“First, this was a retrospective study using medical records. Thus, the surveillance endoscopy intervals were not consistent in all patients and were subjectively determined by the physician in charge, which may have biased the outcome of the analysis.”
Round 2
Reviewer 2 Report
The authors have answered my questions.